# Quarantine Disinfestation of Papaya Mealybug, *Paracoccus marginatus* (Hemiptera: Pseudococcidae) Using Gamma and X-rays Irradiation

**DOI:** 10.3390/insects14080682

**Published:** 2023-08-02

**Authors:** Zi-Jiao Song, Qing-Ying Zhao, Chen Ma, Ran-Ran Chen, Tian-Bi Ma, Zhi-Hong Li, Guo-Ping Zhan

**Affiliations:** 1Institute of Equipment Technology, Chinese Academy of Inspection and Quarantine, Beijing 100123, China; songzijiao0505@163.com (Z.-J.S.); zhaoqy2021@126.com (Q.-Y.Z.); matb1234@126.com (T.-B.M.); 2Department of Plant Biosecurity, College of Plant Protection, China Agricultural University, Beijing 100193, China; 3Department of Entomology, College of Plant Protection, Nanjing Agricultural University, Nanjing 210095, China; 4College of Environmental and Life Sciences, Health, Engineering and Education, Murdoch University, Perth 6150, Australia; 5Division of Plant Quarantine, National Agro-Tech Extension and Service Center, Beijing 100125, China; macfov@hotmail.com (C.M.); chenranran@agri.gov.cn (R.-R.C.)

**Keywords:** phytosanitary irradiation, *Paracoccus marginatus*, dose-response test, large-scale confirmatory test

## Abstract

**Simple Summary:**

The papaya mealybug is a highly polyphagous invasive pest that poses a quarantine threat to tropical and subtropical countries. Infested commodities in international trade should be subject to phytosanitary treatment, and irradiation treatment is recommended to replace methyl bromide fumigation. The irradiation treatments were conducted on the gravid females and 2-, 4-, and 6-day-old eggs of *Paracoccus marginatus* to develop a technical schedule for phytosanitary purpose. The results showed no significant difference among gravid females and 2-, 4-, and 6-day-old eggs after exposure to a range of X-ray radiation (15–105 Gy). Moreover, the minimum radiation dose for providing quarantine security was estimated and validated in the preliminary and confirmatory tests. In the gamma radiation confirmatory tests, no nymphs emerged when a total of 60,386 gravid females were exposed to a radiation dose range of 146.8 to 185.0 Gy. Following the standard-setting principles, the largest dose value in confirmatory tests is the minimum threshold for phytosanitary treatment. As a result, we strongly recommend a minimum dose of 185 Gy for the phytosanitary irradiation of papaya mealybug-infested commodities, and the treatment efficacy is not less than 99.9950% at the 95% confidence level.

**Abstract:**

*Paracoccus marginatus* is a highly polyphagous invasive pest that poses a significant quarantine threat to tropical and subtropical countries. Infested commodities in international trade should undergo phytosanitary treatment, and irradiation is recommended as a viable alternative to replace methyl bromide fumigation. Dose-response tests were conducted on the 2-, 4-, and 6-day-old eggs and gravid females of *P*. *marginatus* using the X-ray radiation doses of 15–105 Gy with an interval of 15 Gy. Radiotolerance was compared using ANOVA, fiducial overlapping and lethal dose ratio (LDR) test, resulting in no significant difference among treatments, except for the overall mortality and LDR at LD_90_ (a dose causing 90% mortality at 95% confidence level). The estimated dose for LD_99.9968_ was 176.5–185.2 Gy, which was validated in the confirmatory tests. No nymphs emerged from a total of 60,386 gravid females exposed to a gamma radiation dose range of 146.8–185.0 Gy in the confirmatory tests. The largest dose in confirmatory tests should be the minimum threshold for phytosanitary treatment, consequently, a minimum dose of 185 Gy is recommended for the phytosanitary irradiation treatment of papaya mealybug-infested commodities, ensuring a treatment efficacy of ≥99.9950% at 95% confidence level.

## 1. Introduction

The papaya mealybug, *Paracoccus marginatus* Williams and Granara de Willink (Hemiptera: Pseudococcidae), is native to Central America. Since the 1990s, it has spread rapidly in mainly tropical areas of the Caribbean, islands in the Indian and Pacific Oceans, Africa, and southern Asia. This insect pest has also spread throughout most provinces of southern China and is currently distributed in a total of 62 countries worldwide [1,2,3,4]. A genetic analysis on samples of *P. marginatus* collected across Asian (Cambodia, China, India, Indonesia, Malaysia, and Thailand) and African countries had been conducted. The results of the analysis indicated the presence of a single haplotype, which suggests a very recent invasion of *P. marginatus* in Asia [5,6].

Papaya mealybug is an economic important insect which is estimated to cause more than 75% of economic damage and income loss of more than USD 3009 per ha at the farm level in Kenya [7]. It has a wide range of hosts and has recorded damage to 189 genera in 58 families of plants. The main economic crops include papaya, mango, custard apple, emblica, acerola, jackfruit, banana, guava, pomegranate, Indian date, sapodilla, and cassava [1,4]. Both nymphs and adult females of *P. marginatus* insert their stylets into the plant’s leaf epidermis, fruit skin, or stem to feed on the plant sap. Simultaneously, they inject a toxic substance into the plant, leading to chlorosis, distortion, stunting, early leaf and fruit drop, the production of honeydew, sooty mold, and possibly plant death, causing significant economic losses [1,4,6]. Moreover, this insect can act as a vector for plant viruses, such as *Piper yellow mottle virus* [8].

This mealybug is capable of spreading over long distances, with plants for planting, fruits, vegetables, and cut flowers being the primary potential pathways for the spreading. Due to its potential to cause significant economic losses and spread rapidly, it has been listed as a quarantine pest by various countries, including the United States, Canada, European and Mediterranean Plant Protection Organization (EPPO), Australia, and other countries where it is not yet present. As a result, phytosanitary measures including pest risk analysis, inspection, treatment, and eradication are needed to prevent its spread and to reduce the economic losses [1,6,9].

Control measures have been developed for quarantine disinfestation and field controlling, including chemical (fumigation, spraying/dipping pesticide) and physical (irradiation, cold, heat, mechanical clean) treatments on consignments, eradication, or integrated control [6,10,11]. Currently, the cold treatment and methyl bromide fumigation are a common measure to disinfest regulated pests found in fresh commodities; however, long times are needed for the cold treatment and fumigant is restricted in the use of excepted quarantine and pre-shipment since it depletes ozone layers. Thereafter, alternative measures should be developed [12,13].

Irradiation treatment has many advantages, e.g., being fast and residue-free, high penetration in the commodities, effective for a wide range of arthropod pests, and no negative effects that could endanger consumers [14,15]. It is an optimum alternative to methyl bromide fumigation in disinfesting insect pests of fruits and vegetables to overcome quarantine barriers in international trade [16,17]. Phytosanitary irradiation (PI) treatment has been developed and used in international trade since the establishment of the International Standard on Phytosanitary Measure (ISPM) No. 18 (*Guidelines*/*Requirements for the use of irradiation as a phytosanitary measure*) by the secretariat of the International Plant Protection Convention (IPPC) in 2003 (revised in 2023), and the international shipment trials on irradiated mangoes between Australia and New Zealand in 2004 [16,17,18]. Meanwhile, it is crucial to develop appropriate schedules/standards for the application of PI treatment since mealybugs are the second most significantly regulated pests besides fruit flies affecting fruits and vegetables, but there are only two published ISPMs related to mealybugs [1,19,20,21]. Additionally, an approved topic for a generic standard: 2017-012 (*Irradiation treatment for all stages of the family Pseudococcidae* (*generic*)) had to be suspended due to insufficient data [22]. More mealybug species are thereby needed to be investigated to push forward the setting process of the topic.

For the PI treatment of the papaya mealybug, radiotolerance grows with increasing stages and times in the gamma radiation dose-response tests. The LD_99.9_ (a dose leading to 99.9% mortality or prevention at a 95% confidence level (CL)) of 165 and 258 Gy were estimated by the linear regression model to induce lethality in developing stages and sterility in adults, respectively [23]. Unfortunately, LD_99.9968_ prevention of F_1_ generation neonates emergence was neither estimated nor validated by the large-scale confirmatory tests, even though it is essential for the development of an ISPM according to ISPM 28 (*phytosanitary treatments for regulated pests*) [19,20,24]. Therefore, normalized dose-response and large-scale confirmatory tests are required to be performed to determine the phytosanitary treatment dose for the most radio-tolerant stage (gravid females) of *P. marginatus*, and to develop a treatment schedule, national standard, and an IPPC standard (annex to ISPM 28) based on the outcomes of this study.

## 2. Materials and Methods

### 2.1. Insect Rearing

The colony of *P. marginatus* used in this study was collected from papaya and cassava plants (leaves or papaya fruits) in three locations: Pingxiang city, Guangxi Autonomous Region; Guangzhou city; and Shenzhen city, Guangdong Province, China. Mealybugs were reared at the Laboratory of Phytosanitary Treatment and Equipment, Chinese Academy of Inspection and Quarantine, Beijing, China, on insecticide-free sprouting potato tubers, *Solanum tuberosum* L. (Tubiflorae: Solanaceae). The infested tubers were placed in medium-sized plastic boxes (22.5 cm × 15.5 cm × 7.5 cm) with a meshed ventilation window on the lid. The rearing took place in either a constant climate chamber (Binder KBF 720, BINDER GmbH, Tuttlingen, Germany) or a designated rearing room. The conditions for rearing were maintained at a temperature range of 25–28 °C, relative humidity of 60–80%, and a photoperiod of L:D = 12:12 h [25].

To ensure diversity and vitality, the progeny was replaced with field populations every 9 to 12 months, and all species identifications were performed in the technical center of Pingxiang Customs. During the study, the lifespan of *P. marginatus* reared under laboratory conditions was investigated. The durations were found to be 7–9 d, 6–10 d, 5–7 d, and 4–6 d for the egg stage, the 1st, 2nd, and 3rd instar nymph stage (developed to females), respectively. It took approximately 7 days for females to become gravid, and the entire lifespan of the adult female was observed to be 16–25 days [25].

### 2.2. Dose-Response Test Using X-ray Irradiation

#### 2.2.1. X-ray Irradiator

Dose-response tests were performed on eggs and gravid females (containing immature eggs) using an RS-2000 Pro X-ray irradiator (Rad Source Technologies, Inc., Coral Springs, FL, USA). To provide a uniform dose distribution, a reflector (43 cm wide, 38 cm deep, and 43 cm high) is placed 40 cm from the X-ray source at the bottom of the exposure chamber. A Petri dish containing eggs or adult females is placed in the center of the chamber to be irradiated and a dosimeter (Model 2086, RadCal Corp., Monrovia, CA, USA) with a 10 × 6-6 ion beam chamber is placed next to the dish to monitor the absorbed doses. The procedure and designs were similar to the X-ray irradiation of *Dysmicoccus lepelleyi* Betrem [26] and the aerial root mealybug *Pseudococcus baliteus* Lit [15], while the operating parameters were 200 keV and 17.6 mA during the irradiation treatments.

#### 2.2.2. Preparation of Adult Females and Eggs

The adult stage is the most radio-tolerant life stage, and thus gravid females were used as the target stage for PI treatment [14,23], whereas egg-hatching or emergence of F_1_ generation 2nd instar nymphs were used for efficacy evaluation [25,26,27]. Eggs of *P. marginatus* are present in ovisac and are always attached to the body of an adult female, which needs to be carefully removed using a soft brush prior to irradiation treatment.

Preparation of adult females. The gravid females (30–35-day-old) were collected from sprouting potato tubers and then transferred to a 5 cm polystyrene Petri dish with a moist filter paper on the bottom after removing the attached ovisacs and eggs; every 10 females were placed in a dish (as a treatment batch), which was then sealed with a plastic film punched with pinholes for ventilation.

Preparation of eggs. About 20 gravid adult females reared on sprouting potato tubers were initially cleared of any attached ovisacs and eggs. Subsequently, they were transferred into a 9 cm Petri dish containing a layer of moist filter paper on the bottom for laying eggs. After 24 h, the females were returned to the tubers for continuous rearing, and the number of newborn eggs in ovisacs was counted under a stereomicroscope (SteREO Discovery V12, Carl ZEISS, Oberkochen, Germany). Subsequently, the counted eggs were moved to a new Petri dish with moist filter paper on the bottom (120 eggs/dish). The Petri dishes were then returned to the rearing room, allowing the eggs to develop for 2, 4, or 6 days before undergoing treatment. This time span was chosen as eggs typically hatched between 7 and 9 days after laying under the experimental conditions.

#### 2.2.3. X-ray Irradiation Treatment

The Petri dishes containing adults or eggs were subjected to irradiation at the target dose of 0 (as control), 15, 30, 45, 60, 75, 90, and 105 Gy, each dose was replicated three times. The cumulative irradiation dose was recorded from the dosimeter. When half of the target dose was reached, the irradiator was temporarily paused, and the door was opened to rotate the Petri dishes 180° horizontally or reposition them manually to achieve a better dose uniformity. The dose rate monitored in this testing was approximately 6.1–6.2 Gy/min. In addition to the treated eggs and adults, all the control eggs (120 eggs/dish) and adults (10 gravid adult females as a batch) were brought into the irradiation room but received no treatment.

#### 2.2.4. Bioassays after Irradiation Treatment

After the X-ray irradiation treatment, the 2-, 4-, and 6-day-old eggs were returned to the rearing room for hatching (occurred 7–9 days after laying). Any unhatched eggs were subsequently examined under a stereomicroscope 20 days after laying. The irradiated adult females were transferred to new sprouting potato tubers in a separate box to continue their development. The newborn eggs and ovisacs were collected and counted every 2 days, and then they were reared in the rearing room. After 20 days from their laying, any unhatched eggs were checked.

### 2.3. Large-Scale Confirmatory Trials Using Gamma Radiation

#### 2.3.1. Gamma Radiation Facility and Treatment

Large-scale confirmatory tests are essential to validate the phytosanitary treatment dose, which was initially estimated through the dose-response test, resulting in a specific treatment efficacy (e.g., LD_99.99_ at 95% CL). These tests are crucial for developing a technical schedule or standard for PI treatment. To achieve this efficacy, a minimum of 30,000 of the most tolerant stage individuals (gravid females) were subjected to treatment following the guidelines of ISPM 18 and 28 [16,17,18,19,28]. The confirmatory tests on *P. marginatus* gravid females were conducted using a Cobalt-60 gamma irradiator provided by the National Institute of Metrology Research in Beijing, China. The Fricke system was used for dose-mapping and monitoring the absorbed dose [29].

To obtain gravid females, parent gravid females were injected onto sprouting potato tubers. The tubers were placed in plastic boxes of medium size weighing between 35 and 70 g, with a diameter ranging from 3 to 4.5 cm and a length of 4.5 to 6 cm. After 4 days, the injected parent females were removed, and the infested tubers were kept in a rearing room for approximately 30 days to allow them to develop into gravid females. Prior to irradiation, a total of 439 and 817 infested tubers were prepared for the first and second irradiation treatments, respectively. These tubers were carefully placed and secured in boxes to prevent movement during transportation, with approximately 15 tubers in each box. About 10% of boxes were left untreated as a control group. Subsequently, all the boxes were packed into cardboard containers and transported to the gamma irradiator.

During the irradiation process, the boxes were positioned 100 cm away from the center of radiation source and exposed to a target dose of 165 Gy. At mid-exposure, the irradiator was temporarily halted, and the plastic boxes were rotated 180° horizontally to achieve a more uniform dose distribution. To monitor dose variations, fifteen Fricke dosimeters were utilized for each irradiation treatment.

#### 2.3.2. Post-Treatment Rearing and Bioassays

After irradiation, the potato tubers were packed into cardboard containers again and transported back to the laboratory. They were kept separately from the control group to prevent cross-infestation. Both the irradiated and controlled mealybugs were maintained under the same rearing conditions mentioned above.

To continue rearing the papaya mealybugs, the gravid females (distinguished by body size and color) on each tuber were counted, and the tubers, along with the mealybugs were then transferred to large-sized plastic boxes (34.0 cm × 23.5 cm × 14.0 cm) with three meshed ventilation windows on the lid. Approximately 2/5 of new potato tubers were added to allow for any F_1_ generation nymphs to feed. Eggs laid by irradiated or untreated females were collected in Petri dishes containing moist filter paper at the bottom every 4 days until the female died. In the control group, the hatch rate for the eggs was checked by counting all the initial number of eggs and the unhatched number after 2 weeks. For the eggs laid by irradiated females, they were placed in the center of the Petri dish, and a layer of sticky shellac was applied around the eggs to collect the hatched 1st instar nymphs.

In the large-scale confirmatory tests, the number of eggs was estimated based on the number of gravid females multiplied by 255, which was the mean count obtained from a total of 720 females treated in the dose-response tests.

### 2.4. Statistical Analysis

To determine the effects of radiation dose on different ages of eggs, and to compare radiotolerance among 0-, 2-, 4-, and 6-day-old eggs, mortality data in the X-ray irradiation dose-response tests were subjected to two-way ANOVA after correcting with Abbot’s formula [30]. Means (±SD, for all mortality) were compared by Tukey’s multiple comparison tests using DPS software [31].

For the probit analysis of the dose-response data, the PoloPlus 2.0 program was used to estimate the minimum absorbed dose leading to 90% (LD_90_), 99% (LD_99_), or 99.9968% (LD_99.9968_, probit-9) mortality, along with their corresponding confidence intervals (CI) at 95% CL. The data used in the estimation include any dose causing <100% mortality and the lowest dose leading to 100% mortality [32,33].

To assess the significance of tolerance, overlapping tests were conducted on the 95% CI at LD_90_, LD_99_, and LD_99.9968_. The results revealed no significant difference between females and developing eggs. Subsequently, pairwise comparison tests were performed by calculating the 95% CI of the lethal dose ratios (LDRs) at LD_90_, LD_99_, and LD_99.9968_, respectively. The criterion for significance was set in order that 1 should not be included in the 95% CI [33,34].

For the confirmatory tests, the treatment efficacy (1 − *Pu*) at a specific CL was calculated using Equation (1) when no nymphs emerge from the treated gravid female:(1)$$1-Pu={\left(1-C\right)}^{1/n}$$
where Pu is the acceptable level of survivorship (normally 0.01% or 0.0032%), *C* is the CL, and *n* is the number of treated females [28,32]. Normally, the mortality proportion (1 − *Pu*) at 95% CL is used, and the value is calculated according to the number of irradiated *P. marginatus* gravid females (≥30,000 for PI treatment) [12,35].

## 3. Results

### 3.1. Dose-Response Test

#### 3.1.1. Impact on Egg Number Laid by Irradiated Females

Mean (±SE) number of eggs laid by irradiated gravid females of *P. marginatus* is listed in Table 1. Results of irradiation effect derived from two-way ANOVA showed that the number of eggs was not affected by the radiation dose (*F* = 1.33, df = 7.95, *p* = 0.2490) or the interaction of dose × age (*F* = 0.36, df = 14.72, *p* = 0.9821). However, it was observed that only age significantly influenced the main effect (*F* = 139.54, df = 2.95, *p* ˂ 0.0001).

There were no significant differences in the number of eggs for the first 4 days, but a substantial decrease was observed from 5 to 6 days, suggesting that no significant irradiation effect was observed when the papaya mealybugs were exposed to radiation doses below 105 Gy. In addition, the total number of eggs laid per female within 6 days was recorded as 255.6 ± 61.8.

#### 3.1.2. Two-Way ANOVA on Dose Mortality of X-ray Irradiation

When *P. marginatus* eggs (including gravid females containing immature eggs) were exposed to X-ray irradiation, the mortality of eggs generally increased with the increasing dose within an age, and the minimum doses causing 100% mortality for 0-, 2-, 4-, and 6-day-old eggs were >105 Gy, 105 Gy, 105 Gy, and >105 Gy, respectively (Table 2). To investigate the significant difference of egg mortality, the dose-mortality data were subjected to two-way ANOVA, which showed that the effects on mortality of *P. marginatus* were significant for both age (*F* = 9.25, df = 3.83, *p* ≤ 0.0001) and dose (*F* = 44.29, df = 6.83, *p* ≤ 0.0001), but not significant for the age × dose interaction effects (*F* = 0.55, df = 18.56, *p* = 0.9171). Therefore, the main effects can be analyzed independently.

For the effect of radiation dose, the mortality of eggs increased significantly with the increasing dose within the same age (Table 2). Regarding the effect of age, there were no significant differences within the same radiation dose, except for 30 Gy and 45 Gy, where the 0-day-old eggs (16.9 ± 5.0%, 33.8 ± 9.5%) obtained the lowest mortality, followed by the 2-day-old (43.6 ± 15.3%, 57.1 ± 18.2%) and 6-day-old eggs (28.1 ± 14.8%, 61.5 ± 9.9%). The mortality of the 4-day-old eggs (48.8 ± 6.5%, 70.0 ± 19.0%) was significantly higher compared to the other groups (Table 2).

With regard to comparing the overall mortality, the mean value for 0-day-old eggs (50.0 ± 32.0%) was significantly lower than that for other ages, indicating that it is the most tolerant life stage. However, no significant difference was observed among the 2-day-old (68.6 ± 28.7%), 4-day-old (71.9 ± 27.8%), and 6-day-old eggs (65.4 ± 30.2%), suggesting no significant difference in radiotolerance among these ages.

#### 3.1.3. Probit Analysis of Dose-Mortality Data

Results derived from the probit analysis, using the probit model on the dose-mortality data of *P. marginatus* gravid females and eggs, are listed in Table 3. These results include the estimated valve for LD_90_, LD_99_, and LD_99.9968_ and their corresponding CI at 95% CL, as well as information on heterogeneity, slope, intercept, and the comparison of the lethal dose ratio (LDR) (Table 4). It is worth noting that the radiation dose was not log-transformed, which is the default setting for probit analysis. Furthermore, the very high heterogeneity value (79.4, Table 3) presented in this study may be attributed to the huge number of 0-day-old eggs and significant experimental errors in the dose-response tests.

However, the 2-, 4-, and 6-day-old eggs showed very similar slopes and intercepts, indicating similar radiotolerance. On the other hand, the 0-day-old eggs obtained a larger slope value, while having the lowest intercept, suggesting a higher resistance to radiation (Table 3). Therefore, to compare radiotolerance effectively, other measures, such as CI overlap and LDR testing, should be employed.

Among different ages of the eggs, all the 95% CI for LD_90_, LD_99_, and LD_998.9968_ overlapped with each other, indicating no significant radiotolerance difference. However, the larger range of CI in produced eggs could be attributed to larger experimental errors. When pairwise comparison of LDR (Table 4) was used, 0-day-old eggs achieved the largest slope (0.034) and LD_90_, indicating the most resistant age among the eggs. The produced eggs showed similar tolerance. Therefore, the results in radiotolerance derived from the probit analysis were consistent with those from the two-way ANOVA (Table 2).

Regarding the comparison of LD_99_, 0-day-old and 6-day-old eggs obtained similar values, significantly larger than those for 2-day-old and 4-day-old eggs, resulting in a stronger radiotolerance. However, there was no significant difference in LD_99.9968_, even though it is an extrapolated value, illustrating that no radiotolerance difference exists in all the eggs.

#### 3.1.4. Comprehensive Comparison of Radiotolerance

To elucidate the irregular sequence of radiotolerance and provide insights into the radiotolerance of *P. marginatus* eggs, all the raw mortality, expected dose-mortality curves, and dose-probit lines are shown in Figure 1 and Figure 2. The 0-day-old eggs displayed notably lower mortality compared to the 6-day-old eggs, followed by 2- and 4-day-old eggs, indicating a higher resistance level than the produced eggs. Consequently, the 0-day-old eggs exhibited significant lower mortality when subjected to ANOVA (Table 2) and significant radiotolerance in the LDR test at LD_90_ (Table 3 and Table 4). This observation could potentially be attributed to the lack of 100% mortality data and the observed lower mortality at the lower doses of 15 and 30 Gy for the 0- and 6-day-old eggs [32], as illustrated in Figure 1.

The expected dose-probit lines for 2-, 4-, and 6-day-old eggs demonstrated a roughly parallel relationship. In contrast, the dose-probit line for 0-day-old eggs appeared in a lower position initially and displayed a steeper slope (0.034 vs. 0.026–0.028 for produced eggs) [32,35]. Notably, no intersections were observed at probit-5.0 (LD_50_), probit-6.28 (line A, LD_90_), and even at probit-7.32 (line B, LD_99_). However, the line intersected with the lines for 2- and 4-day-old eggs at approximately probit-8.20 (line C, LD_99.932_), and the intersection continued until probit-9 (line D, LD_99.9968_) with the line for 6-day-old eggs (Figure 2).

Overall, the 0-day-old eggs exhibited a noticeable level of radiotolerance below LD_99.932_; beyond this point, they demonstrated a similar level of tolerance to the produced eggs of *P. marginatus*. The produced egg may replace gravid female for PI dose-response test.

### 3.2. Large-Scale Confirmatory Tests

For phytosanitary treatment, the criteria for efficacy evaluation are normally mortality or prevention of development/reproduction at probit-9 level, and the estimated probit-9 value is ordinarily used as the target dose for the large-scale confirmatory tests [10,12,15,27], then, we can select eggs at different ages for conducting the confirmatory tests with target dose range from 161 to 199 Gy (Table 3, Figure 2). In the large-scale confirmatory tests, a dose of 165 Gy was used as the target dose, the actual absorbed doses measured by Fricke dosimeters were ranged from 150.5 Gy to 183.6 Gy and 146.8 Gy to 185.0 Gy at the dose rate of 1.26 and 1.23 Gy/min for the first and second trails, respectively, which resulted in the dose uniformity ratio (DUR, the ratio of the maximum dose divided by the minimum dose) of 1.22 and 1.26, respectively for these two tests (Table 5).

There was no F_1_ generation egg hatched, and thus no F_1_ crawler emerged from an estimated 60,386 irradiated gravid females reared on sprouting potatoes (Table 5). In addition, no dead females were found in the radiation treatment to the oviposition period, indicating that the number of gravid females used for the calculation was not adjusted [12,35], whereas the mortality of eggs in the control group was 2.84% and 3.13% for these two confirmatory tests, indicating that the F_1_ generation developed normally in the controls (Table 5). Therefore, efficacy for gamma irradiation calculated by Equation (1) is 99.9950% at 95% CL. Since the maximum radiation dose in the confirmatory tests should be the minimum threshold for PI treatment, the phytosanitary radiation dose (minimum dose) should be not less than 185 Gy.

## 4. Discussion

In this research, the produced eggs at 2-, 4-, and 6-day-old and immature eggs within gravid female of *P. marginatus* were exposed to X-ray irradiation at dose of 15–105 Gy to compare the rate for egg-hatching and assess any potential difference in radiotolerance. Statistical analyses, including ANOVA (Table 2), CI overlapping (Table 3), and LDR test (Table 3 and Table 4) were employed [11,15,32], revealing no significant differences in radiotolerance among the groups. This result agrees with the relative radiotolerance of *D. lepelleyi* eggs irradiated by X-rays, in which the immature eggs seem to be more tolerant than 1-, 2-, and 3-day-old eggs [26]. This also indicates that the egg of papaya mealybug developed well within the female body, and the reason for this phenomenon requires further investigation. On the contrary, radiotolerance tends to increase with the developmental age of eggs for other mealybugs such as *Ps. baliteus* where 0-day-old eggs (immature eggs within the gravid females) were significantly more sensitive to radiation than eggs at 2-, 4-, and 6-day-old [15], and *Ps. comstocki* Kuwana [36]. These observations align with the general trend observed in arthropods, where radiotolerance typically increases with the developmental stage and time when using a common criterion [14].

In order to establish a technical schedule/standard for PI treatment of regulated insect pests, conducting dose-response tests to predict the treatment intensity of large-scale confirmatory trials becomes a critical and logical step. These tests help in determining the most tolerant stage(s) of the insects, which will then be subjected to large-scale confirmatory trials using an estimated dose (target dose) [15,18,24]. In previous studies involving various mealybugs and scale insects, such as *Aonidiella aurantia* Maskell [37], *Aspidiotus destructor* Signoret [38,39], *Hemiberlesia lataniae* Signoret [40], *Pl. citri* Risso and *Pl*. *ficus* Signoret [41], *Pl. lilacinus* [27], *Ps. baliteus* [15], and *Ps. jackbeardsleyi* Gimpel & Miller [42,43], the target dose for large-scale testing was obtained from probit analysis on the most tolerant stage(s), usually the gravid females.

For *P. marginatus*, however, the estimated probit-9 value was 176.7 Gy (160.7–198.9 Gy, 95% CL) for gravid females, and 178.5, 182.9, and 185.2 Gy (95% CI: 143.8–272.1 Gy) for the produced eggs (Table 3). Consequently, a target dose of 165 Gy, the lower CI of probit-9 for gravid female, was used for the confirmatory tests [12,15,43,44]. Notably, no F_1_ generation neonate developed from an estimation of 60,386 gravid females reared on potato tubers and their immature eggs (≥1.5 billion) (Table 5), resulting in a treatment efficacy ≥99.9950% at 95% CL. The results demonstrated that no F_1_ generation neonate developed at this estimated dose, validating the efficacy of the probit-9 estimation [28,32]. Moreover, according to the standard-setting principles, the largest dose value in confirmatory tests should be the minimum threshold for phytosanitary treatment [16,17,19,24]. Regarding the largest radiation dose of 185 Gy monitored in the confirmatory tests, which is equal to or greater than the estimated mean values of probit-9 for the gravid females and 2–6-day-old eggs (Table 3), it can be recommended as the phytosanitary treatment doses for commodities in international trade [14,20,21,45]. Regarding the PI treatment, the currently available X-rays, gamma rays, and electron beams have the same biological effect under equal doses. Therefore, according to the provisions of ISPM 18, any of these radiation types can be used for phytosanitary treatment of papaya mealybug-infested commodities [14,17,18].

In probit analysis, the slopes, intercepts, and heterogeneity values play a crucial role in quantifying the relationship between the response and predictors, providing baseline probabilities, and assessing the goodness of fit of the model to the data [11,31,32,35]. Therefore, these parameters were used to comprehend the dose-response relationship, such as the observed similar radiotolerance among produced eggs and the large heterogeneity leading to a significant width of the CI with overlapping value (Table 3, Figure 2). Additionally, the estimated LD_99.9968_ value for produced eggs (Table 3) will be increased from 4.22- to 5.33-fold with a wide range of CI if the radiation dose was logarithmically transformed during the probit analysis, and the calculation for adult female was suspended due to heterogeneity factor exceeding 100 [32]. These findings are in agreement with similar results obtained in the PI treatment of the cacao mealybug *Planococcus lilacinus* Cockerell [27]. Therefore, the traditional logarithmic transformation of treatment dose is deemed unsuitable for dose-response data analysis in this investigation.

Ionizing radiation, when used for PI treatment, does not result in significant acute mortality at the applied doses. However, it prevents the development and/or reproduction [11,14,15,18,19,20]. In contrast to other phytosanitary treatments like fumigation or cold treatment, the efficacy for PI treatment is not measured based on acute mortality, since acute mortality is not required to prevent the establishment of a pest, and the doses needed to achieve 100% acute mortality are higher than the largest tolerance threshold for most fresh commodities [12,14,17,46]. In order to facilitate the application of PI treatment of *P. marginatus*, we prefer to choose the effect of radiation on preventing the emergence of 2nd instar nymphs as the end point for efficacy evaluation, rather than neonate (1st instar nymphs) from the irradiated females, even though the oviparous mealybugs lay eggs, since it is difficult to find out all eggs or neonates lurking in the female’s abdomen before irradiation treatment [19,20,27,37,38]. Furthermore, very high radiation doses are required to prevent hatching of ready-to-hatch eggs, for instance, the minimum dose to prevent egg-hatching of *Pl. minor* Maskell is different from 7–14-day-old and un-oviposited eggs, which are >250 Gy and 150 Gy, respectively [47].

For PI studies on oviparous mealybugs, preventing egg-hatching of the F_1_ generation could be used as the efficacy criterion, thereby using eggs to replace females in the dose-response tests to determine the relative tolerance and estimate the probit-9 mortality value [15,26,35]. During the life cycle of mealybugs, immature eggs in the body of gravid females act as the connector between parent females and F_1_ generation eggs (for oviparous reproduction type) or neonates (deuterotokous ovoviviparous reproduction type). Generally, radiotolerance in arthropod increases with the developmental stages and times when a common criterion is used. The most developed stage, the gravid female of *P. marginatus*, is the most tolerant stage and should be used as the target stage to be tested in the PI researches according to ISPM 18 and 28 [14,16,17,23]. Then, the produced eggs, same as the eggs of *D. lepelleyi* [27] and *Ps. baliteus* [15], can be used as alternatives to gravid females for conducting the dose-response test.

## Figures and Tables

**Figure 1 insects-14-00682-f001:**
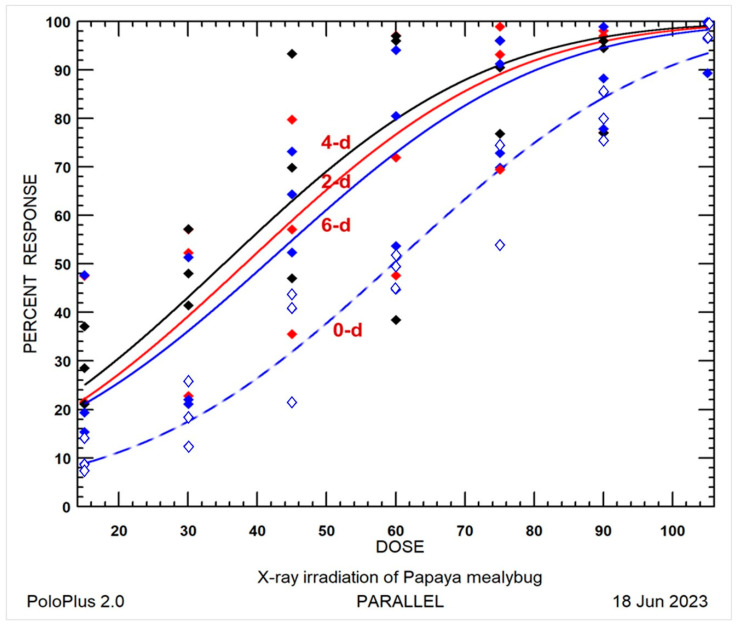
Probit model expected dose-percent mortality curves for 0-, 2-, 4-, and 6-day-old eggs of *Paracoccus marginatus* irradiated at dose of 15–105 Gy with intervals of 15 Gy: (**- -**◇) 0-day-old eggs, (**—**◆) 6-day-old eggs, (**—**◆) 2-day-old eggs, and (**—**◆) 4-day-old eggs.

**Figure 2 insects-14-00682-f002:**
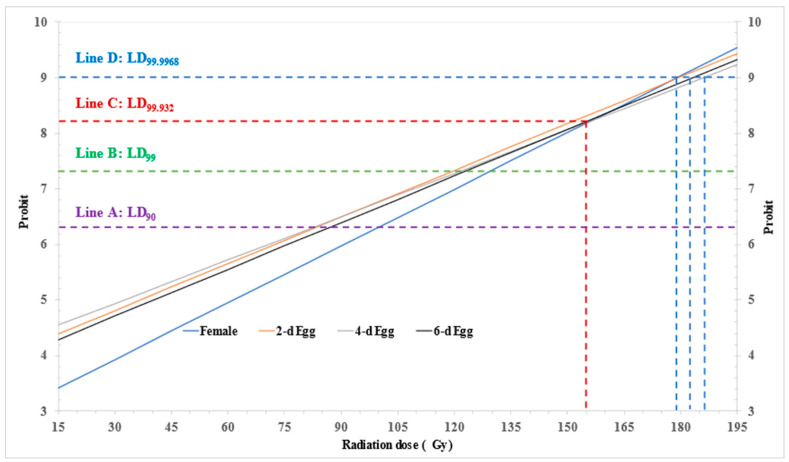
Probit model expected dose-probit lines for 0-, 2-, 4-, and 6-day-old eggs of *Paracoccus marginatus* irradiated at dose of 15–105 Gy with intervals of 15 Gy.

**Table 1 insects-14-00682-t001:** Result of two-way ANOVA on the number (mean ± SE, %) of *Paracoccus marginatus* eggs laid by females irradiated at different doses.

Ages	Mean Number (±SE) of Eggs Laid by Irradiated Females ^a^							
	0 Gy	15 Gy	30 Gy	45 Gy	60 Gy	75 Gy	90 Gy	105 Gy
0–2 d	1305.0 ± 98.1 aA	1339.3 ± 191.9 aA	1250.5 ± 151.8 aA	1163.8 ± 132.0 aA	1457.0 ± 97.4 aA	998.8 ± 74.0 aA	1255.5 ± 107.2 aA	1281.8 ± 237.5 aA
3–4 d	1139.8 ± 186.0 aA	1311.5 ± 322.9 aA	1215.8 ± 340.9 aA	982.0 ± 86.2 aA	1246.5 ± 108.1 aA	869.8 ± 29.0 aA	1145.5 ± 108.7 aA	1228.8 ± 211.5 aA
5–6 d	189.0 ± 11.0 aB	199.8 ± 10.5 aB	186.5 ± 11.5 aB	208.0 ± 9.1 aB	198.8 ± 13.0 aB	124.0 ± 4.7 aB	95.8 ± 8.9 aB	60.5 ± 7.0 aB

^a^ Means followed by different upper-case letters (for age within a column) and lower-case letters (for dose within a row) are significantly different (*p* < 0.05; Tukey test).

**Table 2 insects-14-00682-t002:** Result of two-way ANOVA on the corrected mortality (mean ± SD, %) of *Paracoccus marginatus* eggs irradiated with X-rays at different doses.

Ages	Corrected Mortality (%) of Eggs Irradiated at Dose of ^a^						
	15 Gy	30 Gy	45 Gy	60 Gy	75 Gy	90 Gy	105 Gy
0 d	7.9 ± 2.5 eA	16.9 ± 5.0 deB	33.8 ± 9.5 cdeB	47.5 ± 2.6 bcdA	65.4 ± 8.7 abcA	79.9 ± 4.0 abA	98.6 ± 1.3 aA
2 d	29.6 ± 12.4 dA	43.6 ± 15.3 cdAB	57.1 ± 18.2 bcdAB	72.0 ± 20.4 abcA	87.2 ± 12.9 abA	90.7 ± 9.7 abA	100.0 ± 0.0 aA
4 d	28.7 ± 6.5 cA	48.8 ± 6.5 bcA	70.0 ± 19.0 abA	77.2 ± 27.5 abA	87.9 ± 8.1 aA	90.7 ± 6.5 aA	100.0 ± 0.0 aA
6 d	23.9 ± 15.1 cA	28.1 ± 14.8 bcAB	61.5 ± 9.9 abAB	75.0 ± 17.6 aA	86.1 ± 10.5 aA	87.8 ± 9.1 aA	95.1 ± 4.6 aA

^a^ Means (±SD, %) followed by different upper-case letters (for each age within a column) and lower-case letters (for dose within a row) are significantly different (*p* < 0.05; Tukey test).

**Table 3 insects-14-00682-t003:** Probit analysis on prevention of egg-hatching when the gravid females and 2-,4-,6-day-old eggs of *Paracoccus marginatus* were irradiated with X-rays.

Age	No. of Eggs	Slope ± SE	Intercept ± SE	Estimated LDs and 95% CI (Gy) ^a^			Heterogeneity ^b^
				LD_90_	LD_99_	LD_99.9968_	
0 d	53,249	0.034 ± 0.000	−2.090 ± 0.022	97.8 (92.0–105.4) a	128.2 (118.7–141.0) a	176.7 (160.7–198.9) a	79.4
2 d	2247	0.028 ± 0.001	−1.030 ± 0.069	82.0 (70.5–102.5) bc	119.1 (99.6–157.7) b	178.5 (144.4–248.0) a	14.4
4 d	2095	0.026 ± 0.001	−0.843 ± 0.070	80.2 (67.6–104.7) c	119.7 (97.7–168.2) b	182.9 (143.8–272.1) a	15.5
6 d	2195	0.028 ± 0.001	−1.130 ± 0.077	87.1 (76.0–105.0) b	124.8 (106.4–158.3) ab	185.2 (153.2–245.5) a	11.1

^a^ LD_90_, LD_99_, and LD_99.9968_ at 95% CL of *P. marginatus* eggs are calculated with probit model. Within a column, estimated LD values followed by different letters are significantly different (*p* < 0.05, LDR test). ^b^ Chi-square (χ2) divided by the degree of freedom.

**Table 4 insects-14-00682-t004:** Pairwise comparison of LDR for the differently aged eggs of *Paracoccus marginatus* treated with X-rays irradiation.

Reference Age	Pairwise Age	95% CI of Lethal Dose Ratio		
		LD_90_	LD_99_	LD_99.9968_
0 d	2 d	1.25 × 10^12^–∞	273.36–4.24 × 10^15^	0.00–7.55 × 10^9^
	4 d	4.49 × 10^13^–∞	13.80–6.31 × 10^13^	0.00–1.18 × 10^7^
	6 d	7.89 × 10^6^–4.78 × 10^14^	0.00–1.73 × 10^10^	0.00–4.59 × 10^3^
2 d	4 d	0.00–1.38 × 10^7^	0.00–1.44 × 10^9^	0.00–1.17 × 10^13^
	6 d	0.00–1.86	0.00–5.28 × 10^3^	0.00–8.80 × 10^9^
4 d	6 d	0.00–0.04	0.00–6.54 × 10^4^	0.00–2.79 × 10^15^

**Table 5 insects-14-00682-t005:** Results of the large-scale confirmatory tests of *Paracoccus marginatus* adult females reared on sprouting potatoes using gamma irradiation.

Irradiation Date	Absorbed Dose (Gy)	DUR	No. of Potatoes	No. of Females	F_1_ Generation Eggs	
					No. ^a^	Mortality (%)
13 Febuary 2023	150.4–183.6	1.22	439	9992	2.5 × 10^6^	100
control	0	-	60	2056	5.2 × 10^5^	2.84
10 April 2023	146.8–185.0	1.26	817	50,376	1.3 × 10^7^	100
control	0	-	75	4749	1.2 × 10^6^	3.13

^a^ Total number of eggs was estimated using 255 eggs per female as the average value, which was obtained from a total of 720 females in Table 1.

## Data Availability

All data presented in this study are available in the article.

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
