# Peer review of "Quarantine Disinfestation of Papaya Mealybug, Paracoccus marginatus (Hemiptera: Pseudococcidae) Using Gamma and X-rays Irradiation"

_insects, 2023, doi:10.3390/insects14080682_

Round 1

Reviewer 1 Report

This manuscript described the quarantine disinfection of papaya mealybug using Gamma and X-rays irradiation. You investigated radiotolerance, the minimum radiation dose for providing quarantine security and the presence of nymphs.

The manuscript is clear and relevant for the field. 

 I have three remarks concerning the manuscript:

- You used X-ray irradiation treatment but you made your large-scale confirmatory trials using Gamma ray.  Are you not afraid to create a bias?

-Captions for table 1 & 2 are not very clear. You speak about upper-case and lower-case but what is “a” and “B” and “A”. For me, it is necessary to enlight the captions.

- In the summary and in the introduction, you are mentioning methyl bromide fumigation and cold treatment. According to me, in the discussion, you must discuss the effectiveness of irradiation versus these methods actually employed.

Author Response

Dear Editor and Reviewers,

I extend my heartfelt gratitude to you and the reviewers for providing such insightful comments that have been instrumental in enhancing the quality of this manuscript. Enclosed in the following pages, you will find a meticulous point-by-point response addressing each of the reviewers' comments, with all the corresponding revisions clearly highlighted in red.

I wish to affirm that all authors have diligently participated in editing and proofreading this manuscript.

Thank you for your time and consideration. We eagerly await your response.

Sincerely,

Guoping Zhan

Review 1 (Round 1)

Comments and Suggestions for Authors

This manuscript described the quarantine disinfection of papaya mealybug using Gamma and X-rays irradiation. You investigated radiotolerance, the minimum radiation dose for providing quarantine security and the presence of nymphs.

The manuscript is clear and relevant for the field. 

 I have three remarks concerning the manuscript:

Revisions 1. You used X-ray irradiation treatment but you made your large-scale confirmatory trials using Gamma ray.  Are you not afraid to create a bias?

Response: Thank you for bringing this to our attention. We have now included a sentence in lines 420-423 to clarify that both types of rays have the same biological effect.

Revisions 2. Captions for table 1 & 2 are not very clear. You speak about upper-case and lower-case but what is “a” and “B” and “A”. For me, it is necessary to enlight the captions.

Response: The “a” and “A” are respectively the lower-case and upper-case letter in my mining, and we have made the necessary corrections to ensure consistent usage throughout. Regarding the captions, we will adhere to the style of the magazine for enlightenment and ensure they are appropriately formatted and in line with the publication's guidelines.

Revisions 3. In the summary and in the introduction, you are mentioning methyl bromide fumigation and cold treatment. According to me, in the discussion, you must discuss the effectiveness of irradiation versus these methods actually employed.

Response: Agree. We make changes in Line 81-82, and add discussion in Line 437-443.

Reviewer 2 Report

There is potential for further edits, particularly in the introduction, methodology, and discussion sections, to improve readability and make it more concise. 

There is potential for further edits, particularly in the introduction, methodology, and discussion sections, to improve readability and make it more concise. 

Author Response

Dear Editor and Reviewers,

I extend my heartfelt gratitude to you and the reviewers for providing such insightful comments that have been instrumental in enhancing the quality of this manuscript. Enclosed in the following pages, you will find a meticulous point-by-point response addressing each of the reviewers' comments, with all the corresponding revisions clearly highlighted in red. Please see the attachment.

I wish to affirm that all authors have diligently participated in editing and proofreading this manuscript.

Thank you for your time and consideration. We eagerly await your response.

Sincerely,

Guoping Zhan

Review 2 (Round 1)

Comments and Suggestions for Authors

General comments: There is potential for further edits, particularly in the introduction, methodology, and discussion sections, to improve readability and make it more concise. 

Response: Agree. We have thoroughly revised the entire manuscript, carefully incorporating your comments and suggestions. The necessary changes have been highlighted in red to ensure transparency and clarity. We sincerely hope that this revision meets your expectations, and we eagerly await any further comments or suggestions you may have. Your valuable input has been instrumental in improving the quality of the paper, and we appreciate your guidance. Thank you for your time and consideration.

Revisions 1. Title: L.2: Add Genus and species names before the Order and Family names

Response: Added.

Simple Summary:

Revisions 2. L16-423: Be consistent with the comma (,) usage when listing three or more words or items. Example L.16: …….invasive, polyphagous pest, and poses….OR……invasive, polyphagous pest and poses….

Response: We acknowledge your feedback, and we have diligently reviewed and corrected all instances of comma usage in the manuscript.

Revisions 3. L16-423: Semi column (;) was used wrongly throughout the text; I suggest the authors use transition words or "Full stop" more frequently instead of the semi column.

Response: Agree. We acknowledge the need for proper column usage and have now diligently corrected all instances in the manuscript.

Revisions 4. L.17-18: Rearrange to read "Infested commodities should undergo phytosanitary treatment internationally, while irradiation is recommended to replace methyl bromide fumigation."

Response: Agree. We make changes in line 18-20.

Revisions 5. L.19: Add "a" before technical

Response: Added.

Revisions 6. L.19: Change to "conducted on…...gravid females of P. marginatus and the 2-, 4-, and 6-d old eggs to develop technical protocol for phytosanitary purpose.’’

Response: Changed. Thanks.

Revisions 7. L.20-26: Change to " The results showed no significant difference among gravid females, 2-, 4-, and 6-d old eggs of P. marginatus after exposure to a range of X-ray radiation (15-105 Gy). Also, the minimum radiation dose effects on mealybugs. In the confirmatory test, no nymphs emerged when a total of 60,386 gravid females were exposed to a Gamma-ray radiation dose range of 146.8 to 185.0 Gy.

Response: Changed. Thank you.

Revisions 8. L.27-28: Why should the authors recommend 185.0 Gy for phytosanitary treatment of papaya mealybug-infested commodities when no nymphs emerge after exposure to the lowest Gamma-ray radiation (146.8 Gy) in the confirmatory test? I understand that we mostly use the upper limit of the 95% CI of the LD99.9968 for developing technical standards. Can it be explained in the text to the audience?

Response: we explain the reason in line 357-361, 377-380, and 414-420.

Revisions 9. L.28: Change to ...." Mealybug-infested ...

Response: Changed.

Abstract:

Revisions 10. L.30: As already mentioned stick to one when listing 3 or more items in the text; (A, B, and C) Or (A, B and C)

Response: Revised.

Revisions 11. L.31: Change to "... commodities"

Response: Changed.

Revisions 12. L.35: Remove "respectively".

Response: Done.

Revisions 13. L.36: Change to" resulting in no significant difference among treatments, except...….. (a dose leading to 90%...)). The estimated dose............

Response: Changed from Line 37.

Revisions 14. L.38: Remove "because Gamma ............pests.

Response: Deleted.

Revisions 15. L.39- 43: Rewrite OR replace with the edited Lines of 24-29.

Response: Agree. We make changes in line 39-44.

Introduction:

Revisions 17. L.51: Change"..…southe rn Asia. This insect pest has also… ....."

Response: Agree. We make changes in line 52.

Revisions 18. L.52: Replace "Ahmed et al. 2015" with [5] and do this for other in text citations. Also, change all the superscript of the in text citations

Response: Deleted.

Revisions 19. L.54: Remove";" after African countries

Response: Changed.

Revisions 20. L.58: Add “import” before insect. Change to “Kenya”

Response: Added in line 58, and changed in line 60.

Revisions 21. L.59: 58 families of what?

Response: Agree. We add ‘of plants’ in line 61.

Revisions 22. L.62: "feed"

Response: Change in line 64.

Revisions 23. L.65: Change to "resulting"

Response: Changed in line 66-67.

Revisions 24. L.66-67: Rewrite

Response: Agree. We rewrite it in line 69-70.

Revisions 25. L.82: Remove "effectively"

Response: Removed.

Revisions 26. L.89: Change to "Standards

Response: Changed in line 95.

Revisions 27. L.91: Remove "have been"

Response: Removed.

Revisions 28. L.91-95. Rewrite by removing the II  ;ff

Response: Agree. we make changes in line 98-101.

Revisions 29. L.96: Remove "Seth et al. (2016) and all such in text citations and replace them with their assigned numbers.

Response: Removed.

Materials and Methods

Revisions 30. L.118: Change to "every". How will changing insect cultures affect the dose-response mortality in the context of genetic variations in radio-tolerant in different population? Explanation required in the Discussion section.

Response: Thank you for your understanding. We initially intended to include discussions in the manuscript, but due to the lack of sufficient supporting references, we decided not to include them at this time. We believe it is essential to maintain the academic rigor and integrity of the work.

Revisions 31. L.119: Remove "Dr. Yong Zhong" and write" was conducted in Technical center ….......

Response: Changed in line 128-129.

Revisions 32. L.122:" .....young female to develop.....…

Response: Agree. we make changes from line 132.

Revisions 33. L. 129: change to " A Petri dish"

Response: Changed in line 139.

Revisions 34. L. 137: Which sex male or female adult stage? Specify

Response: We changed to ‘gravid females’ in line 147.

Revisions 35. L. 140: How is the attached eggs removed?

Response: We make changes in line 149-151.

Revisions 36. L.144: Delete "awayII

Response: Deleted. Thanks.

Revisions 37. L.149: Change to "after removing attached ovisacs… ..…II

Response: Agree. We have made the necessary changes in lines 157-158.

Revisions 38. L.160: How is the Petri dishes rotated during irradiation without the experimenter getting exposed? Add this details

Response: Agree. We make changes in from line 170.

Revisions 39. L.177: Delete" Since ........." andstart with "The confirmatory tests............"

Response: Deleted.

Revisions 40. L.181-185: Rewrite

Response: Agree. We make changes from line 196-200.

Revisions 41. L.227: Change to" ....... When no nymphs emerge from the........"

Response: Changed in line 248.

Results.

Revisions 43. L.235: Change to" Mean (+SE) number of eggs laid by irradiated gravid females

Response: Changed in line 257.

Revisions 43. L. 237-240. Replace "duration" by "stage". Duration is mostly used when studying the effect of exposure time in dose response experiments. Also, when interpreting the out of two-way ANOVA (E.g. Factorial experiment), start from the interaction effect (x), however, if there is no significant interactions, interpret the main effects.

Response: Thank you for your notification. We make changes from line 260.

Revisions 44. L.243: Delete "by"

Response: Deleted.

Revisions 45. L.244: Delete" mean+SE

Response: Deleted.

Revisions 46. L.256: Change to F=9.25; df=3,83;

Response: Thank you for your notification. We have changed to the style required by the journal.

Revisions 47. L.258: why Standard Deviation (SD) used? And not SE for Table 2.

Response: We chose 'mean±SD' for analyzing the mortality data and made a note of it in Section 2.4 (Line 234).

Revisions 48. Table 2: The 2nd column should be deleted since it makes the Table footnotes invalid4

Response: Deleted. We add the overall mortality value from line 293.

Revisions 49. Table 3: The 95% CI interval of LD90 for 0-d is incorrect; cross check

Response: Revised. Thanks.

Revisions 50. Table 3: The very high heterogeneity values are typically unacceptable in dose response tests. Is it because cultures were replaced every 9 mo?

Response: Thank you for your notifications. We have provided an explanation for this in lines 303-305. It was mentioned that for the Polo software, heterogeneity values ≥100 are considered unacceptable.

Revisions 51. Table 3: Were the 95% CI for LDs used to establish the significant difference among treatments? If yes, cross check and add the same alphabets for overlapping Cis within columns

Response: We think we used the 95% CI for LDs to present the range of LDs estimated by the software automatically. Additionally, it was utilized for the CI overlapping test to determine the significant difference between two treatments.

Revisions 52. Table 3: The practical significance of the slope, intercepts, and heterogeneity values should be explained in the discussion section and their overall impact is establishing the technical standard or protocol.

Response: Agree. We add explanations in line 424-429.

Revisions 53. Table 4: Can alphabets be assigned to show LDs that were significant as in Table 3. Figure 1. Can the nature of the curves be explained by the heterogeneity values in Table 3 in the Discussion section?

Response: Table 4 was used to explain the significant difference among the LDs in Table 3 during the LDR testing, for example, comparing the significant difference between 0-d and 2-d-old eggs, 95% CI of LDS at LD90 and LD99 exclude 1, but include 1 at LD99.9968, then, significant tolerance presented LD90 and LD99 but none for LD99.9968.

For the Figure 1, large experimental error and huge number of insects used for the estimations resulted in a large heterogeneity value, as we described in line 303-305. Due to the complexity of the statistical issue of heterogeneity, and our limited expertise in this area, we do not possess the capability to discuss it comprehensively. Furthermore, the length of the article has already become quite extensive. Therefore, we kindly request your understanding and allow us to refrain from discussing this matter. Thank you for your understanding!

Discussion:

Revisions 54. L.347-424: Most of these lines should be the Material and Methods (statistical analysis) and Results section. Example L.352-354, L.374-375 etc.

However, the major finding of this study show be discussed along with relevant literature and tell the practical importance of the studies.

Response: Agree. Thank you for your suggestion. We have removed all the lines that were previously used in the sections of materials and methods, as well as results, except for the necessary content relevant to the discussions. By doing so, we aim to streamline the manuscript and ensure a more focused and concise presentation of our findings. If you have any further recommendations or feedback, please feel free to share them with us. Your input is greatly appreciated!

Revisions 55. L.392: Change to 2nd

Response: Changed. Thanks.

Revisions 56. Overall, the Discussion Section needs to be improved

Response: Agree. We have made revisions throughout the discussion to ensure the accuracy and clarity of the content. Our goal is to address any concerns and ensure the manuscript meets your satisfaction. We appreciate your feedback and are committed to delivering a high-quality and well-refined work. If you have any further suggestions or requests, please let us know, and we will be more than happy to accommodate them.

Revisions 57. Reference: L.452: Cross check this citation well.

Response: Done. Thanks.